# Fixed-Seat Rowing versus Sliding-Seat Rowing: Effects on Physical Fitness in Breast Cancer Survivors

**DOI:** 10.3390/cancers16122207

**Published:** 2024-06-13

**Authors:** Juan Gavala-González, Mateo Real-Pérez, Laura Benítez-García, José C. Fernández-García

**Affiliations:** 1Department of Physical Education and Sports, Universidad de Sevilla, 41003 Sevilla, Spain; jgavala@us.es; 2Researching in Sport Science: Research Group (CTS-563) of the Andalusian Research Plan, University of Málaga, 41003 Málaga, Spainjcfg@uma.es (J.C.F.-G.); 3Department of Didactics of Languages, Arts and Sport, University of Málaga, Andalucía-Tech, IBIMA, 29010 Málaga, Spain

**Keywords:** rowing, breast cancer, physical activity, anthropometry, exercise

## Abstract

**Simple Summary:**

Breast cancer is the most common cancer in women, often causing physical side effects like loss of muscle strength, reduced mobility, and fatigue. This study explores the impact of fixed-seat rowing (FSR) and sliding-seat rowing (SSR) on physical fitness and body composition in female breast cancer survivors to identify the more effective exercise modality. Forty participants rowed twice weekly for 24 weeks. The results indicate significant improvements in weight, body mass index, waist and hip circumference, muscle strength, aerobic capacity, and flexibility, with SSR showing better overall outcomes. These findings suggest that SSR may be a more beneficial exercise for breast cancer survivors. This research underscores the value of tailored exercise programs in improving recovery and quality of life for these women, potentially shaping future rehabilitation strategies within the research community.

**Abstract:**

This study aimed to analyze the effects of a team rowing-based training program on physical fitness and anthropometric parameters in female breast cancer survivors (n = 40; 56.78 ± 6.38 years). The participants were divided into two groups: one rowed in fixed-seat rowing (FSR) boats (n = 20; 56.35 ± 4.89 years), and the other rowed in sliding-seat rowing (SSR) boats (n = 20; 57.20 ± 7.7 years). Both groups engaged in two 75 min sessions per week for 24 weeks. Significant improvements were observed in both groups in terms of weight (FSR: −1.93 kg, SSR: −1.75 kg), body mass index (FSR: −0.73 kg/m^2^, SSR: −0.67 kg/m^2^), waist circumference (FSR: −2.83 cm, SSR: −3.66 cm), and hip circumference (FSR: −2.02 cm, SSR: −2.88 cm). Muscle strength improved in the lower extremities (jump test: FSR: 2.99 cm, SSR: 3.11 cm) and upper extremities (dominant: FSR: 4.13 kgf, SSR: 4.34 kgf; non-dominant: FSR: 3.67 kgf, SSR: 3.32 kgf). Aerobic capacity also improved, with the SSR group showing a greater increase (FSR: 63.05 m, SSR: 93.65 m). Flexibility tests revealed better results in the SSR group for both dominant (SSR: 1.75 cm vs. FSR: −5.55 cm) and non-dominant limbs (SSR: 1.72 cm vs. FSR: −3.81 cm). These findings suggest that the type of rowing modality can influence physical fitness outcomes, with the SSR group showing superior improvements compared to the FSR group.

## 1. Introduction

Among the different types of cancer, breast cancer is the most commonly diagnosed cancer in women worldwide, accounting for 2.3 million new cases and about 700,000 deaths in 2020 [1]. Nevertheless, advances in cancer research have effectively improved the early detection of the disease, as well as treatment and relapse prevention, resulting in improved survival rates after surgery [2], and what was previously a fatal disease has become a chronic one.

The most common cancer treatments, such as chemotherapy, radiotherapy, and surgery, have side effects in the short and long term that compromise the physical conditions of the affected women [2]. Some of these negative effects are associated with different current treatments, including loss of muscle mass and strength, reduced mobility in the upper extremities, worsening of aerobic capacity, and other symptoms such as the onset of lymphedema, fatigue, depression, and cardiac toxicity, contributing to a decline in quality of life [1,3,4].

Recognizing the need to find therapies that help alleviate the symptoms associated with treatment in women with breast cancer, recent studies have shown that maintaining adequate levels of physical activity is associated with improved functionality and mobility in breast cancer survivors [5,6,7,8,9,10], thereby improving the side effects of treatments, such as cancer-related fatigue or symptoms of depression [10,11,12,13]. Accordingly, we believe that a training program led by professionals could improve quality of life [8,10] as well as decrease tumor recurrence, increasing the overall breast cancer survival rate [14]. As a result of the preceding research, the American College of Sports Medicine recommends that cancer survivors should perform at least 150 min of moderate physical activity or 75 min of vigorous exercise and incorporate strength training twice a week [15].

In contrast, despite the widely demonstrated benefits of physical activity and exercise on physical function and associated symptomatology in breast cancer survivors [13,16,17,18], most women significantly reduce their physical activity during cancer treatment [19,20], even below their pre-diagnosis levels [18]. Because of this, it is important to promote strategies and activities that encourage adherence to physical activity [21].

Rowing, in its FSR and SSR modalities, has been shown to provide both physical and emotional benefits as a complementary non-drug therapy for women with breast cancer [8,9,10,22]. The inclusion of this sport for promoting complementary therapies aimed at this population stems from its very characteristics, since the action of rowing involves both the muscles of the upper and lower extremities as well as most of the muscles of the body [23]. However, the main difference with other sports lies in the cyclic and alternative action of the flexion–extension movement of the upper and lower limbs, while the muscles of the back and abdomen act to stabilize the technique [24].

When examining the technique in both modalities, it can be seen that FSR uses a stroke characterized by an asymmetrical action of the musculature, requiring compensatory effort [25]. In contrast, SSR involves comprehensive movements, engaging the musculature in almost identical proportions, as the movement is symmetrical in all planes. The position of the body segments a priori may be more suitable for women who have had cancer and undergone upper limb surgeries, since rowing does not require forced movements [9]. The relationship between physical activity and breast cancer has been widely studied in the existing literature [1,5,16,26], but in the case of rowing, it is virtually nonexistent [9,22]. Therefore, the importance of our study is to analyze the impact of an FSR- and SSR-based training program on physical fitness and body composition in female breast cancer survivors.

## 2. Materials and Methods

The aim of this study is to determine which form of exercise and which boat (FSR boat vs. SSR boat) is better for improving the physical fitness of female breast cancer survivors in order to recommend a specific type of physical therapy.

### 2.1. Study Design

This study is a part of parallel clinical trials, and for this purpose, a 24-week, twice-weekly training program was designed that could be carried out in both FSR and SSR boats (Figure 1).

On the other hand, several associations of women with breast cancer were contacted and offered the possibility of a free 24-week rowing program and a physical assessment before and at the end of the program. The only requirements for participation were to have suffered from breast cancer and to have the approval of their oncologist for moderate physical activity.

Before initiating the training program, a meeting was held to explain the nature of the study, the objectives, and the commitments involved. The participants (Figure 2) signed an informed consent form, and the project coordinator explained the details of the study, which follows the ethical principles for research with human subjects of the Declaration of Helsinki [27] and the ethical considerations of Sport and Exercise Science Research [28]. This study was registered and approved by the Ethics Committee of the University of Malaga with the number 65-2020-H. All of the information included in the research was collected following the Organic Law 3/2018 of 5 December on Personal Data Protection and guarantee of digital rights, regarding the protection of personal data of Spanish legislation.

### 2.2. Participants

Subjects (N = 40) aged 56.78 + 6.38 years were recruited, provided they had survived breast cancer for 6.58 + 5.72 years, with the characteristics shown in Table 1, and had the approval of their oncologists to perform physical activity.

### 2.3. Data Records

Height was measured with an SECA model 213 portable stadiometer (Seca GmbH & Co. KG, Hamburg, Germany), using the Frankfurt plane for head positioning. For weight, a Tanita BC 545N (Tanita Corporation, Tokyo, Japan) scale was used, following the manufacturer’s protocol for both clothing and previous liquid or food intake.

To measure physical fitness, the tests and procedure described by Gavala et al. (2020) were used [9]. The tests used to measure strength were the counter movement jump (CMJ) for the lower extremities (with the My Jump 2 application; accuracy of ±1 cm) and the handgrip test to measure hand and forearm muscle strength with Takei 5401 dynamometer (Takei Scientific Instruments Co., Ltd., Niigata, Japan); accuracy of ±2 kgf. Cardiac endurance was measured using the 6 min walk test. For overall flexibility, the sit-and-reach test was used (with the Baseline Sit ‘n Reach measuring box; accuracy of ±1 cm). Finally, a Cescorf tape measure (Cescorf, Porto Alegre, Brazil) with an accuracy of ±1 mm was used to measure circumferences.

### 2.4. Procedure

Of all the women who showed interest, a series of physical fitness tests were carried out and, with equal anthropometric parameters (weight, height, BMI) and test results, one was assigned to one group (FSR or SSR), and the one with the most similar results was assigned to the other group (SSR or FSR). The sample was divided into two training groups with similar measurements (Table 2):

The SSR group performed a rowing program in sliding-seat boats, while the FSR group performed the same training in fixed-seat boats, with both groups being homogeneous.

During the first week of the study, different measurements were taken. To carry out the comparative study, a training protocol was implemented for a period of 6 months, where two 75 min sessions per week were held for two groups of women who had never participated in this sport.

After the measurements were taken, a 24-week program was completed, supervised by trainers who ensured attendance (participants who did not reach 90% participation were excluded), correct execution of the movements, and adequate intensity according to the training cycle. The exercise program was divided into three 8-week phases (Table 3). These stages are marked by progressive increases in technical difficulty, distance covered, and intensity. For this purpose, the trainers regulated the effort of the rowers by using the subjective perception of effort (Börg scale) [29].

### 2.5. Statistical Analysis

All analyses were performed with the IBM SPSS Statistics 25 statistical package. The level of significance was set at *p* < 0.05. A frequency analysis was carried out for the variables of age, height, weight, and body mass index (BMI). The fit of the different variables to the normal distribution was assessed by both graphic procedures and the Shapiro–Wilk test.

To determine the influence of the boat type on the anthropometric and physical fitness variables, a comparative analysis of the means (ANOVA) was performed, taking into account the mean values and the standard deviation of the boat factor with respect to the variables in the measurements obtained in the pre-test. To analyze whether there were differences according to the rowing training performed by the participants, the data from the pre-test and post-test measurements were compared through the different tests. The estimated between-subject marginal means (Boat*Measurement) and the standard deviation were considered when quantifying the interaction between the variables and their longitudinal evolution through a repeated measures ANOVA, applying the post hoc Bonferroni test. In addition, a graphic analysis of the different variables was carried out using box and whisker plots. The analysis of the statistically significant differences in the comparison between groups was made with Student’s *t*-test for independent groups.

## 3. Results

Table 4 displays the means of the study variables, as well as the differences before and after the 6-month FSR and SSR training program. In most cases, we found improvements in the values after the training program in the study subjects. Moreover, these improvements were virtually statistically significant in their entirety, both for the variables associated with body composition and those related to physical condition, except for the non-dominant back scratch test (*p* = 0.258). The critical levels for Cohen effect sizes identified in a *t*-test (d) are 0.20 for minor effects, 0.50 for medium effects, and 0.80 for significant effects.

The following figures compare the changes produced according to the different training protocols. Focusing on body composition, Figure 3 shows a slight improvement in both groups (FSR: Δ_pre-post_ BMI = −0.73/SSR: Δ_pre-post_ BMI = −0.67; *p* = 0.000), although it is slightly higher in the FSR group.

In Figure 4, statistically significant improvements were detected in both waist circumference (FSR: Δ_pre-post_ waist circumference = −2.83/SSR: Δ_pre-post_ waist circumference = −3.66; *p* = 0.000) and hip circumference (FSR: Δ_pre-post_ hip circumference = −2.2/SSR: Δ_pre-post_ hip circumference = −2.88; *p* = 0.000), although the values were slightly higher in the SSR group.

Regarding the parameters associated with flexibility, Figure 5 shows significant improvements in both groups for the general flexibility variable with the sit-and-reach test (FSR: Δ_pre-post_ sit and reach = +3.53; *p* = 0.027/SSR: Δ_pre-post_ sit and reach = +4.4/*p* = 0.027). An analysis of the values obtained in the back scratch test (BST) shows a significant worsening in upper extremity flexibility in the FSR rowers in both the dominant arm (FSR: Δ_pre-post_ dominant BST = −5.55; *p* = 0.02) and the non-dominant arm, although this was not significant (FSR: Δ_pre-post_ non-dominant BST = −3.81; *p* = 0.258). Conversely, data from subjects participating in the SSR training program showed improvements in both the dominant arm (SSR: Δ_pre-post_ dominant BST = +1.75; *p* = 0.02) and the non-dominant arm (SSR: Δ_pre-post_ non-dominant BST = +1.72; *p* = 0.258).

Regarding the muscle strength tests, Figure 6 shows a significant improvement in women who trained in both fixed-seat (FSR: Δ_pre-post_ CMJ = +2.99; *p* = 0.000) and sliding-seat boats (SSR: Δ_pre-post_ CMJ = +3.11; *p* = 0.000), although the results are slightly higher in the latter group.

Figure 7 illustrates the results of the handgrip test with significant improvements in both training groups, both for the dominant arm (FSR: Δ_pre-post_ dominant handgrip = +4.13/SSR: Δ_pre-post_ dominant handgrip = +4.34; *p* = 0.000) and the non-dominant arm (FSR: Δ_pre-post_ non-dominant handgrip = +3.67/SSR: Δ_pre-post_ non-dominant handgrip = +3.32; *p* = 0.000).

Finally, regarding endurance, Figure 8 shows a significant improvement in the SSR training group (SSR: Δ_pre-post_ 6 min walk test = +93.65; *p* = 0.000) and also in the FSR group, but the latter results are somewhat lower (FSR: Δ_pre-post_ 6 min walk test = +63.05; *p* = 0.000).

## 4. Discussion

Several studies [16,30] have concluded that the regular practice of physical activity and its relationship with cancer has multiple benefits at the cellular level, and it can improve the quality of life of patients undergoing multiple drug treatments, reduce the side effects of anti-cancer therapies, and improve prognosis and survival after cancer [10,13,17]. It is therefore important to be aware of the potential benefits for the body from maintaining a healthy lifestyle and to encourage the maintenance of physical activity and exercise levels in a controlled and professionally supervised manner as a complementary non-pharmacological therapy for the treatment of patients with cancer [30].

In this study, it was found that after the intervention program, both the FSR and SSR training groups showed positive and significant changes in the anthropometric and physical fitness variables measured, including muscle strength, aerobic capacity, and general flexibility. Consequently, it can be reported that 6 months of rowing training had positive effects on physical fitness and anthropometric measures in female breast cancer survivors. These results coincide with those of previous studies [8,9,10] showing that rowing can be a safe and effective sport to not only improve the fitness of patients but also their quality of life.

Unlike some currently published studies, such as those by Moro et al. [31], where no statistically significant differences were found after a 12-week dragon boat intervention protocol in variables associated with body composition (BMI and upper limb circumference), we observed significant results in both training groups regarding the obtained anthropometric measures. In this regard, the importance of maintaining adequate BMI levels in cancer is related to the role of insulin and the risk of non-insulin-dependent diabetes, which are strongly influenced by body fat distribution and physical activity levels [16]. Additionally, hyperinsulinemia causes an increase in insulin-like growth factor (IGF)-1 and a decrease in its binding proteins. Specifically, IGF-1 plays a primary role in carcinogenesis [32]. Recent research has revealed that exercise can significantly reduce the postoperative levels of IGF-1 and proteins related to inflammatory processes, such as C-reactive protein (CRP), tumor necrosis factor-alpha (TNFα), and interleukin-6 (IL-6) in adult and elderly women with breast cancer [19]. Consistent with this study’s results, scientific evidence supports that regarding the decrease in the described factors, the duration of training appears to regulate this relationship, as these adaptations are stipulated to occur when the training program consists of more than 12 weeks of exercise [16]. Furthermore, it has been shown that a training program lasting 12 weeks or less only reduced the CRP levels, limiting the potential benefits of physical exercise in this specific population [33].

On the other hand, regarding the influence of exercise on variables associated with the physical conditions of the study participants, scientific evidence demonstrates significant advances in this field of knowledge [1,5,34]. Literature reviews, such as those by Sturgeon et al. [5], indicate that an aerobic exercise program for women with breast cancer can have a moderate effect on improving vascular function. Additionally, aerobic exercise can serve as a protective factor against cancer, as it has been shown to play an important role in promoting angiogenesis and decreasing the expressions of pro-angiogenic markers [19]. These markers are crucial because they are essential for tumor cell growth, dissemination, and metastasis. The results show a significant increase in the distance covered in the 6-min walk test (6MWT) in both the FSR group (+63.05 m) and the SSR group (+93.65 m), which is in line with the values obtained after 12 weeks of fixed bench rowing training [8,31].

Regarding muscular strength, is important to recognize that one of the main issues faced by breast cancer survivors is the loss of muscle mass and functionality, particularly after surgical intervention and especially in the affected limb, which negatively impacts their quality of life [22,35]. There is limited evidence on the influence of strength training in individuals with breast cancer, but it appears that light strength exercises have a positive effect on perceived effort during the rehabilitation process [36]. After the rowing training program, significant improvements were observed in both upper and lower limb muscular strength in both training groups (FSR and SSR).

With this in mind, there are now associations and clubs that promote rowing as a sport that can benefit breast cancer survivors [10]. However, the most suitable type of boat must be determined since, to date, there is no scientific evidence in this regard. By examining the type of boat used in the research on this topic, the first studies involving group rowing in female breast cancer survivors were undertaken using FSR as part of the training protocol [8,9,10,22]. In these studies, the potential effects of rowing training on physical fitness, body composition, and even cardiac function in groups of female breast cancer survivors were explored using FSR boat programs combining sports therapy [8,9,10] and strength and endurance training. These studies have shown that a 12-week intervention program with SSR boats is capable of producing cardiac adaptations [8] and positively influencing the subjects’ aerobic capacity, muscle strength, and body composition [11,20]. Nevertheless, the literature on this topic is still scarce, and more studies are needed to draw conclusions on which types of boat and training produce more beneficial effects in this specific population.

The results of this study show statistically significant differences between the FSR training group and the SSR group, although in both groups, most variables improved. The SSR group showed a greater increase in the variables associated with physical fitness (muscle strength, aerobic capacity, and flexibility) as well as a more marked decrease in the anthropometric measurements (weight; waist and hip circumference), compared to the FSR group. It should also be noted that, while flexibility in the dominant arm increased significantly in the SSR group (also in the non-dominant upper extremity, although not significantly), in the FSR group, the results worsened in both cases after the training period. This may be due to the asymmetrical involvement of the arm muscles in the FSR rowing technique, resulting in an imbalance in upper extremity development.

Based on this evidence, the SSR boat appears to be a better type of vessel for delivering this form of therapy. However, further research is needed to study upper extremity muscle involvement during rowing and its relationship to mobility and functionality in breast cancer survivors who have undergone treatment to manage breast cancer and who are highly affected in this body area.

This study had some limitations, mainly caused by the limitations established for participating in the study. The predefined inclusion criteria and the inherent characteristics of the sport, carried out outdoors and conditioned by weather circumstances, caused a certain experimental attrition among the subjects who participated in the intervention program. On the other hand, being a pioneering study focused on understanding the influence of rowing on physical fitness in female breast cancer survivors, there are certain limitations, such as the absence of a control group, as well as the use of tests like rowing ergometer tests, spiroergometry tests, echocardiography, or gas analysis, which could provide more in-depth results following the intervention protocol.

## 5. Conclusions

Through this pioneering study, it was confirmed that rowing, in both its variants—FSR and SSR—is a safe activity to consider when prescribing physical exercise to female breast cancer survivors. Although both the FSR and SSR training groups showed improvements in anthropometric measurements and physical fitness variables after a 6-month intervention program, the SSR group made greater progress. Nevertheless, future research is needed to test the relationship between rowing technique and boat type and aspects of physical fitness such as upper extremity flexibility in people who have upper limb impairment, such as female breast cancer survivors.

## Figures and Tables

**Figure 1 cancers-16-02207-f001:**
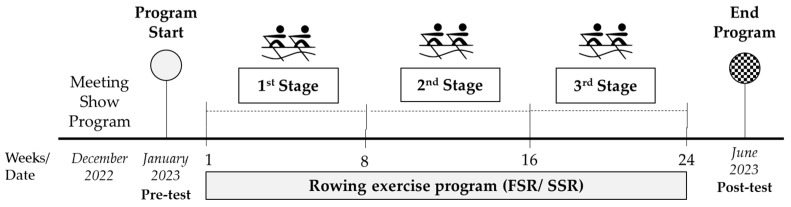
Study’s temporal line.

**Figure 2 cancers-16-02207-f002:**
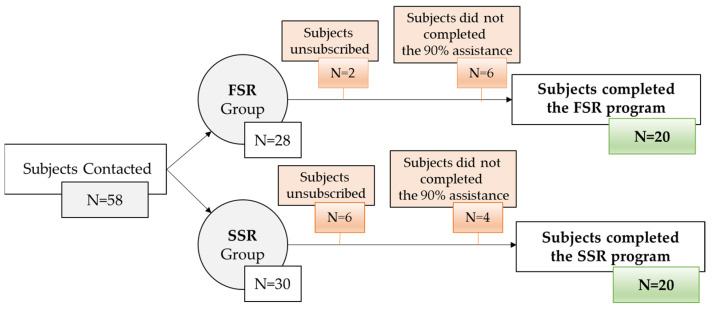
A flow diagram of the sample selected for this study.

**Figure 3 cancers-16-02207-f003:**
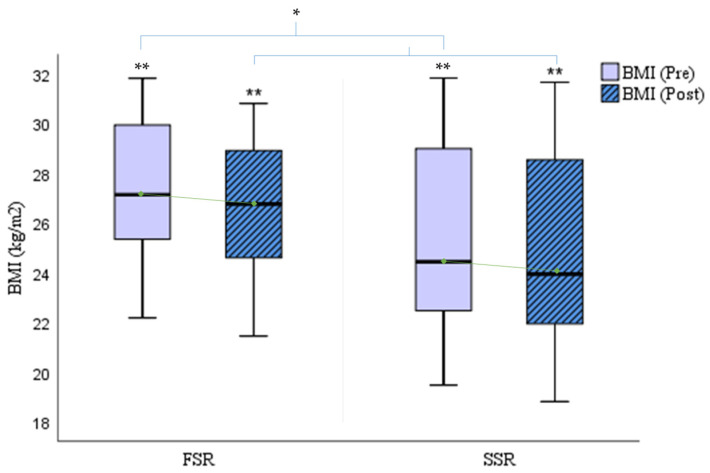
Comparison of BMI (kg/m^2^) pre- and post-rowing training. BMI = Body Mass Index; FSR = Fixed-Seat Rowing; SSR = Sliding-Seat Rowing. * *p* < 0.05; ** *p* < 0.001.

**Figure 4 cancers-16-02207-f004:**
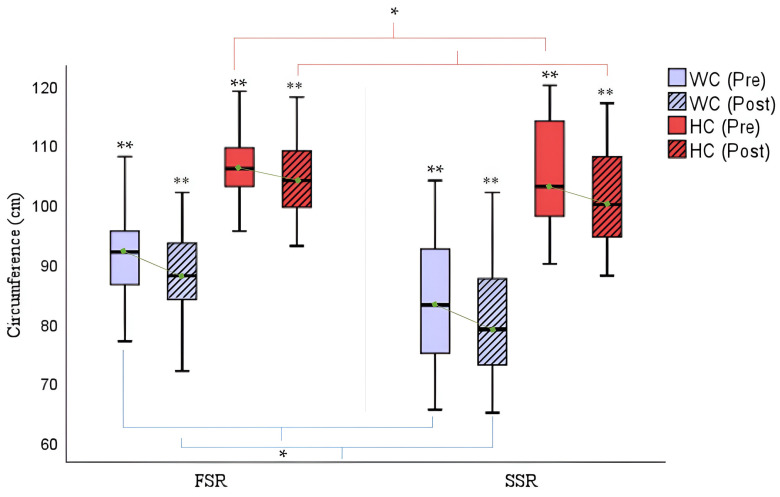
Comparison of circumferences (cm) pre- and post-rowing training. WC = Waist Circumference; HC = Hip Circumference; FSR = Fixed-Seat Rowing; SSR = Sliding-Seat Rowing. * *p* < 0.05; ** *p* < 0.001.

**Figure 5 cancers-16-02207-f005:**
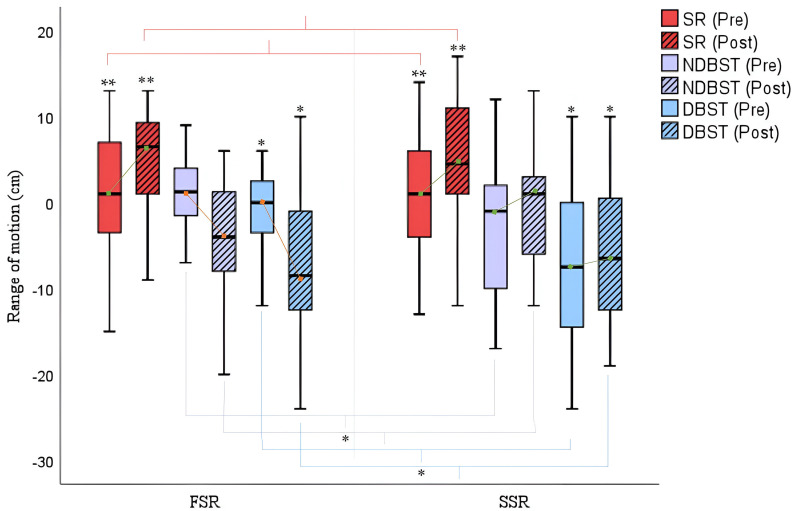
Back scratch test (BST) (cm) pre- and post-rowing training. SR = Sit and Reach; NDBST = Non-Dominant Back Scratch Test; DBST = Dominant Back Scratch Test; FSR = Fixed-Seat Rowing; SSR = Sliding-Seat Rowing. * *p* < 0.05; ** *p* < 0.001.

**Figure 6 cancers-16-02207-f006:**
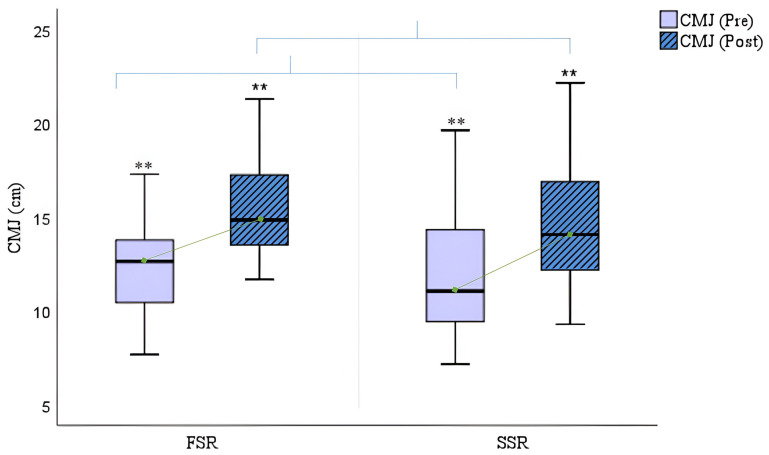
Counter movement jump (CMJ) (cm) pre- and post- rowing training. FSR = Fixed-Seat Rowing; SSR = Sliding-Seat Rowing. ** *p* < 0.001. No statistically significant differences in comparison between groups (Student’s *t*-test for independent groups).

**Figure 7 cancers-16-02207-f007:**
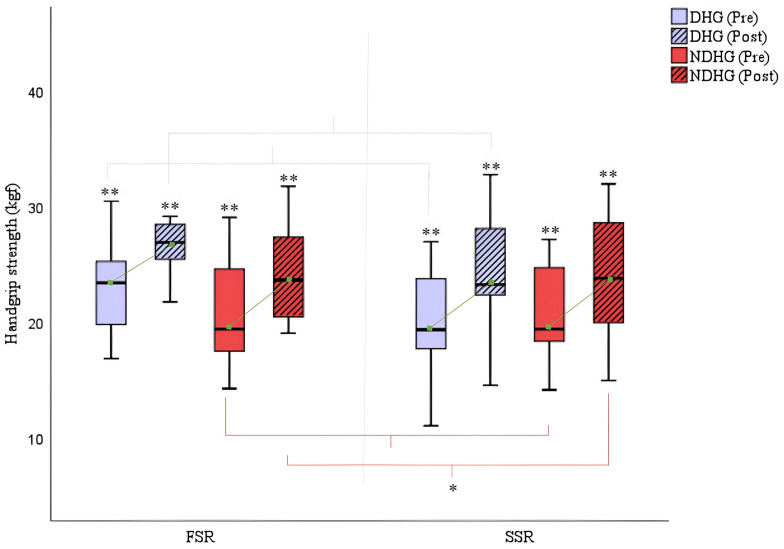
Handgrip (kgf) comparison pre- and post-rowing training. DHG = Dominant Handgrip; NDHG = Non-Dominant Handgrip; FSR = Fixed-Seat Rowing; SSR = Sliding-Seat Rowing. * *p* < 0.05; ** *p* < 0.001.

**Figure 8 cancers-16-02207-f008:**
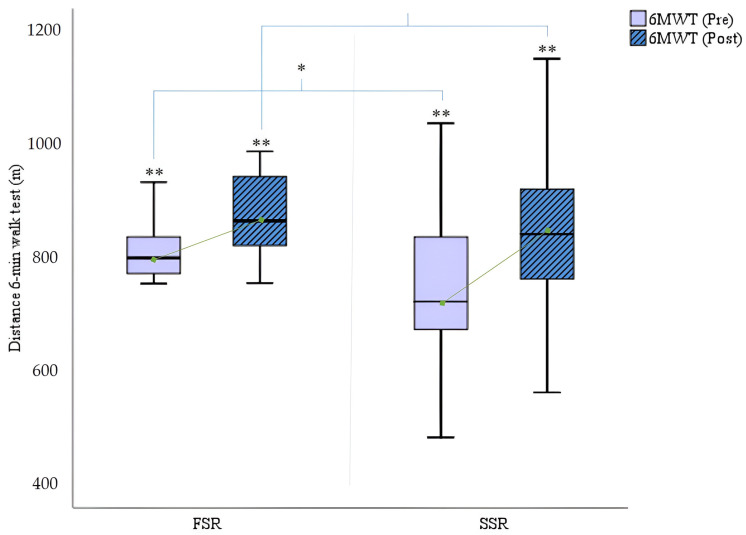
A comparison of the distance in the 6 min walk test (m) pre- and post-rowing training. 6MWT = 6 min walk test; FSR = fixed-seat rowing; SSR = sliding-seat rowing. * *p* < 0.05; ** *p* < 0.001.

**Table 1 cancers-16-02207-t001:** A description of the characteristics of the participating subjects (affected breast, diagnosed stage, and intervention performed).

**Breast (%)**	Right	37.5
	Left	57.5
	Both	5.0
**Stage (%)**	I	7.5
	II	37.5
	III	40.0
	IV	15.0
**Surgery (%)**	Preservation	50.0
	Total Mastectomy	42.5
	Double Mastectomy	7.5

**Table 2 cancers-16-02207-t002:** Descriptive analysis of study subjects according to training group.

	TOTAL (SSR + FSR)	Fixed-Seat Rowing (FSR)	Sliding-Seat Rowing (SSR)	Difference of Means (FSR-SSR)	*p*
Age (years) (SD)	56.78 (6.38)	56.35 (4.89)	57.20 (7.7)	−0.85 (2.04)	0.679
Height (cm) (SD)	162.05 (5.59)	161.90 (4.91)	162.20 (6.33)	−0.3 (1.79)	0.868
Weight (kg) (SD)	69.49 (9.8)	72.05 (8.11)	66.92 (10.85)	3.22 (3.11)	0.307
BMI (kg/m^2^) (SD)	26.48 (3.58)	27.48 (2.73)	25.47 (4.09)	1.50 (1.18)	0.215

**Table 3 cancers-16-02207-t003:** Exercise prescription design for program.

Stage/Duration	Content	Timming	Börg Scale
1st (8 weeks)	Initial phase with mobility exercises, proprioceptive exercises, and postural control exercises.	8 min	5–6
	Main phase with rowing training.	60 min	
	Final phase with stretching.	7 min	
2nd (8 weeks)	Initial phase with mobility exercises, proprioceptive exercises, and postural control exercises.	8 min	6–7
	Main phase with rowing training.	60 min	
	Final phase with stretching.	7 min	
3rd (8 weeks)	Initial phase with mobility exercises, proprioceptive exercises, and postural control exercises.	8 min	7–8
	Main phase with rowing training.	60 min	
	Final phase with stretching.	7 min	

**Table 4 cancers-16-02207-t004:** Between-subject analysis of study variables according to boat type.

	Fixed-Seat Rowing (FSR)			Sliding-Seat Rowing (SSR)			Interaction Effect Boat Measurement			Effect Size
	Pre-Test (SD)	Post-Test (SD)	ΔPre-Post	Pre-Test (SD)	Post-Test (SD)	ΔPre-Post	MS	F	*p*	
**Body composition**										
Weight (kg)	72.05 (8.11)	70.12 (7.88)	−1.93	66.92 (10.85)	65.17 (10.43)	−1.75	263.17	8.87	0.000 **	0.64
BMI (kg/m^2^)	27.48 (2.73)	26.75 (2.66)	−0.73	25.47 (4.09)	24.8 (3.89)	−0.67	40.49	3.34	0.000 **	0.71
Waist circumference (cm)	91.1 (7.3)	88.27 (7.28)	−2.83	84.53 (11.52)	80.87 (10.35)	−3.66	430.99	4.63	0.000 **	0.47
Hip circumference (cm)	106.22 (6.35)	104.2 (6.87)	−2.02	104.45 (9.57)	101.57 (8.89)	−2.88	31.51	0.48	0.000 **	0.51
**Strength**										
CMJ (cm)	12.55 (3.09)	15.54 (2.94)	+2.99	11.71 (3.49)	14.82 (3.65)	+3.11	7.09	0.65	0.000 **	0.62
Dominant handgrip (kgf)	22.69 (3.64)	26.82 (3.87)	+4.13	20.49 (6.21)	24.83 (6.03)	+4.34	48.16	1.85	0.000 **	0.8
Non-dominant handgrip (kgf)	20.55 (4.53)	24.22 (3.96)	+3.67	21.17 (5.61)	24.49 (6.54)	+3.32	3.84	0.15	0.000 **	0.97
**Aerobic capacity**										
6 min walk test (m)	817.25 (51.1)	880.3 (69.96)	+63.05	752.3 (127.37)	845.95 (132.58)	+93.65	42,185.02	4.25	0.000 **	0.67
**Flexibility**										
Sit-and-Reach (cm)	1.40 (6.97)	4.93 (6.05)	+3.53	1.05 (7.49)	5.45 (6.92)	+4.4	1.22	0.02	0.027 *	0.35
Dominant back scratch test (cm)	−1.75 (6.36)	−7.30 (8.03)	−5.55	−7.68 (9.13)	−5.93 (7.55)	+1.75	351.06	5.39	0.02 *	0.22
Non-dominant back scratch test (cm)	0.32 (6.11)	−4.13 (6.76)	−3.81	−2.80 (7.18)	−1.08 (6.23)	+1.72	97.66	2.08	0.258	0.25

Interaction Effect Boat*Measurement refers to the influence of the type of boat used according to the training group on the variables; * *p* < 0.05; ** *p* < 0.001.

## Data Availability

The Ethical Committee of the University of Malaga, which authorized this study, precludes ceding, showing, or disclosing to third parties the database of this research by any means (personal data, medical history, interviews, physical condition tests, and others) because it is considered confidential and sensitive information.

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
