# Peer review of "Fixed-Seat Rowing versus Sliding-Seat Rowing: Effects on Physical Fitness in Breast Cancer Survivors"

_cancers, 2024, doi:10.3390/cancers16122207_

Round 1
Reviewer 1 Report
Comments and Suggestions for Authors
Review report for manuscript "Fixed Seat Rowing versus Sliding Seat Rowing: Effects on Physical Fitness in Breast Cancer Survivors" (cancers-3038726)
The study was conducted on an interesting and promising topic; however, there are significant concerns that cannot be scientifically ignored. The following general comments and suggestions are intended to improve the quality of the manuscript for future attempts.
The physiological fact of the dominance of rowing over other physical exercises, and also the fact that these two rowing exercises (i.e. FSR and SSR) are the most important rowing exercises, should be clearly explained in the intro, supported by valid references (preferably not self-referencing).
While proofreading is highly necessary, many statements should be paraphrased using more advanced scientific grammar and expressions with an integrated passive voice writing style.
The Methods section needs an additional sub-section explaining study design, recruitment of participants, and inclusion/exclusion criteria.
Given the multifaceted nature of study design, the methods section also needs a flowchart or diagram showing sample size, study procedures, timing, exclusion/inclusion criteria and study groups.
Table 1 should be redesigned using a standard table design, including rows and columns.
The presentation of the data in Table 2 is not appropriate and the rows and columns should be reversed. More importantly, the statistical difference between the two groups for each variable should be reported and, if significant, discussed as a potential confounding factor.
The duration of each phase should be reported within Table 3.
The authors should provide information on the confounding variable that may affect the results, preferably in the study design, otherwise as a limitation of the study.
It is not clear whether the p-value reported in Table 4 refers to the examination of the difference between which 2 numbers; please specify as a footnote to the table.
In Table 4, the FSR group showed better results for some variables (e.g. BMI). Therefore, a considerable part of the results and their interpretation should be revised accordingly, e.g. "although slightly higher in the SSR group" (in lines 163-163) is not true.
In all figures, the statistical difference between the two groups (not for each group separately) should be clearly stated. Also, "*p<.05" is not applicable for some figures and should be removed.
The Discussion does not adequately compare the results of the study with the results of comparable or semi-comparable studies. The limitations of the study should be listed and explained more realistically. The Discussion also lacks some parts, such as practical implications and suggestions for future investigations at the end.
Minor points:
Line 21: Longitudinal study is not applicable to this study!
Lines 88-89: The statement is redundant and should be removed.
Comments on the Quality of English LanguageWhile proofreading is necessary, many statements should be paraphrased using more advanced scientific grammar and expressions with an integrated passive voice writing style.
Reviewer 2 Report
Comments and Suggestions for Authors
The abstract needs to be modified. Need for this research methodology has to be more specified. The research gap is to be analyzed. The paper is offshoot of twenty six week boat roaring exercise of the patients. The clinical finding are good but more statistical analysis is required apart from ANOVA analysis. The discussion section needs improvement. The references are adequate. The conclusion is not reflecting the objectives of the paper. The conclusion may be improved based on the revised version of the article. In figure 6 selection of p -value has to brought out.
Comments on the Quality of English Language
NIL
Round 2
Reviewer 1 Report
Comments and Suggestions for Authors
I appreciate authors' efforts in the revisions, which have improved the quality of the report. However, a significant number of concerns remain unaddressed.
Section 2.1 needs structural changes and, more importantly, the addition of further information, e.g. on the type of study, the sample size at different stages (including invitation, initially enrolled, excluded, withdrawn, final sample) and the timing of the study stages (year, months). The statement in lines 92-94 should be referenced. The manuscript needs a flowchart or diagram giving a general overview of the methodological details.
Statistical between-group differences (not within-group differences) should be clearly specified in all figures.
There are many other limitations that have not been specified. Authors could review comparable studies (with similar designs, but perhaps different exercise interventions) for potential limitations in such studies.
The Discussion is too short (compared to the large amount of data presented in the Results), so it needs to cover all key findings, along with additional comparisons with similar studies done in the target population, even with different exercise modalities.
Comments on the Quality of English LanguageThere are still a few minor issues to be resolved, particularly in terms of clarity.
Author Response
Dear Reviewer,
First of all, we would like to once again thank you for your time and dedication to our article, which has undoubtedly improved qualitatively since the first submission. We have added more information in section 2.1 (study type, sample size at different stages, study chronology, and included two graphs that show the entire study development) (lines 86-110).
Regarding the statistics, the analysis of the statistically significant differences in the comparison between groups was made with the Student's t-test for independent groups, and we have presented the results in the graphs. If you need more information, we can provide you with the table containing the results obtained in the group comparison.
We have added study limitations after consulting similar literature and have deepened the discussion. In addition to what was already added in the first round of review, a new, more comprehensive comparative analysis has now been developed (lines 227-263 and 297-301).
Thank you so much for your time, we hope that changes will adapt to the article's needs.
MR.

Reviewer 2 Report
Comments and Suggestions for Authors
All the corrections are included in the paper. Hence, there is no need for further review.
Comments on the Quality of English LanguageNil
Author Response
Dear Reviewer,
First of all, we would like to once again thank you for your time and dedication to our article, which has undoubtedly improved qualitatively since the first submission. We have added more information in section 2.1 (study type, sample size at different stages, study chronology, and included two graphs that show the entire study development) (lines 86-110).
Regarding the statistics, the analysis of the statistically significant differences in the comparison between groups was made with the Student's t-test for independent groups, and we have presented the results in the graphs. If you need more information, we can provide you with the table containing the results obtained in the group comparison.
We have added study limitations after consulting similar literature and have deepened the discussion. In addition to what was already added in the first round of review, a new, more comprehensive comparative analysis has now been developed (lines 227-263 and 297-301).
Round 3
Reviewer 1 Report
Comments and Suggestions for Authors
Well done!
Comments on the Quality of English LanguageIt is almost acceptable!